# PACMANN: EFFICIENT PRIVATE APPROXIMATE NEAREST NEIGHBOR SEARCH

**Mingxun Zhou, Elaine Shi & Giulia Fanti**
Carnegie Mellon University
Pittsburgh, PA 15213, USA
{`mingxunz,rshi,gfanti`}@andrew.cmu.edu

## ABSTRACT

We propose a new private Approximate Nearest Neighbor (ANN) search scheme named PACMANN that allows a client to perform ANN search in a vector database without revealing the query vector to the server. Unlike prior constructions that run encrypted search on the server side, PACMANN carefully offloads limited computation and storage to the client, no longer requiring computationally-intensive cryptographic techniques. Specifically, clients run a graph-based ANN search, where in each hop on the graph, the client privately retrieves local graph information from the server. To make this efficient, we combine two ideas: (1) we adapt a leading graph-based ANN search algorithm to be compatible with private information retrieval (PIR) for subgraph retrieval; (2) we use a recent class of PIR schemes that trade offline preprocessing for online computational efficiency. PACMANN achieves significantly better search quality than the state-of-the-art private ANN search schemes, showing up to $2.5\times$ better search accuracy on real-world datasets than prior work and reaching 90% quality of a state-of-the-art non-private ANN algorithm. Moreover on large datasets with up to 100 million vectors, PACMANN shows better scalability than prior private ANN schemes with up to 62% reduction in computation time and 22% reduction in overall latency.

## 1 INTRODUCTION

Today, most search systems require clients to send a plaintext query to the server, which performs a search over the database and returns the most relevant document(s). This architecture poses a serious privacy risk to users, whose search queries can leak privacy-sensitive information (aol, 2006). To this end, an important problem is that of *private search*: algorithms that allow a user to search a database without revealing their query to the server in plaintext. To date, there have been several proposed private search algorithms, which provide cryptographic privacy guarantees over the user's query (Henzinger et al., 2023; Asi et al., 2024; Servan-Schreiber et al., 2022).[1] However, **existing algorithms suffer from poor tradeoffs between: (1) search quality and/or (2) efficiency.** For example, consider Tiptoe (Henzinger et al., 2023), the state-of-the-art private search algorithm with cryptographically strong privacy guarantees. Tiptoe incurs a linear computation cost for the server per query; that is,

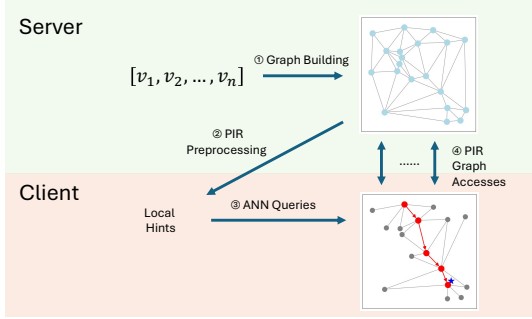

Figure 1: The high-level overview of PACMANN. 1) The server builds a graph structure on the database vectors. 2) The client and the server run the preprocessing protocol for the PIR scheme, storing the local hint in client's storage. 3) The client makes ANN queries. 4) The client runs the graph traversal algorithm locally, but uses the PIR scheme to access the graph information remotely.

---

[1]Cryptographic privacy means that the server learns cryptographically negligible information about the query.

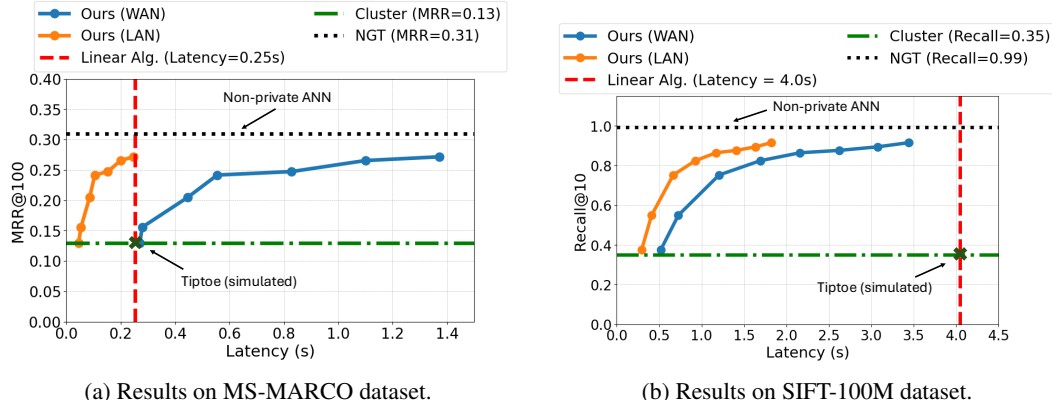

(a) Results on MS-MARCO dataset.        (b) Results on SIFT-100M dataset.

Figure 2: Tradeoff between search quality and latency. "Linear Alg." is an SIMD-optimized linear algorithm that lower bounds the latency of Tiptoe (Henzinger et al., 2023), the state-of-the-art private search algorithm. As a safe lower bound, we do not include network latency for "Linear Alg.". "Cluster" is a clustering-based algorithm that upper bounds the quality of Tiptoe. The intersection of the two can be viewed as our (simulated) result for Tiptoe. We upper bound search quality with NGT, a state-of-the-art non-private ANN search algorithm (Iwasaki & Miyazaki, 2018). We plot PACMANN result in both the LAN (5ms RTT) and WAN (50ms RTT) settings.

the server needs to at least scan through the entire database for each query. Although Tiptoe is parallelizable, the linear computation cost may be a bottleneck for large-scale applications. At the same time, Tiptoe's search accuracy is limited. For example, it only achieves around $40\%$ of the non-private search accuracy in the MS-MARCO dataset.

Typical private search algorithms can be categorized by two main design choices: the search algorithm and the privacy primitive. These two are closely intertwined: the search algorithm must be chosen to be compatible with the privacy primitive, while also ensuring that information is not leaked to the server and the search quality is not overly degraded. For example, Tiptoe makes two design choices that limit its performance: (1) It uses a clustering-based approximate nearest neighbor (ANN) algorithm that significantly limits the search quality. (2) Its privacy guarantees are based on a preprocessed somewhat homomorphic encryption (SHE) scheme, which requires significant computation linear in the dataset size.

In this work, we present PACMANN (Privately ACcess More Approximate Nearest Neighbors), a fully private nearest neighbor search algorithm that addresses the search quality-efficiency tradeoff. Instead of performing encrypted search on the server's side, the client can run the search algorithm locally. To achieve this without the client storing the whole index structure, the client in PACMANN dynamically fetches necessary information from the server. Doing so allows us to use more sophisticated graph-based ANN search algorithms that have been popularized in applications like retrieval-augmented generation (RAG) (Malkov & Yashunin, 2018; Jayaram Subramanya et al., 2019; Iwasaki & Miyazaki, 2018), without designing complicated cryptographic protocols. PACMANN achieves strong privacy with the help of a recently popularized cryptographic primitive called client-preprocessing private information retrieval (PIR) (Zhou et al., 2024).

**Contributions.** In this work, we make two main contributions.

1. **Algorithm Design.** We present the design of PACMANN, which builds on a customized graph-based ANN search algorithms. In graph-based ANN search algorithms (Malkov & Yashunin, 2018; Jayaram Subramanya et al., 2019; Iwasaki & Miyazaki, 2018), each vertex in the graph represents a database vector and is connected to multiple other vertices based on proximity in a vector (embedding ) space. Given a query vector, the algorithms usually start from a given vertex and traverse the graph to find ANNs. In each step of the traversal, the algorithms examine all connected vertices to the current vertex and move to the next vertex that is closer to the query, until a stopping criterion is met. To implement graph traversal algorithm privately, PACMANN requires the client to *locally* run graph traversal over carefully-selected subgraphs. To efficiently retrieve

the appropriate subgraph, we modify an existing, practically-efficient PIR scheme (Zhou et al., 2024) to handle batched queries. Figure 1 gives a high-level system overview of PACMANN.

2. **Empirical results.** We empirically evaluate the performance of PACMANN in terms of search quality, query latency, communication, and storage costs. Our results show that PACMANN achieves significantly better search quality than the clustering-based private ANN search algorithm used by state-of-the-art Tiptoe (Henzinger et al., 2023) and Wally (Asi et al., 2024). For example, our implementation finds the most relevant result in around 63% of the queries on the MSMARCO dataset, compared to 29% for clustering-based algorithms—a **2.1x improvement** in search success rate. In the 100M SIFT dataset, our evaluation shows a **2.5x better** recall@10 compared to Tiptoe. Figure 2 shows that PACMANN achieves 90% of the search quality of the leading non-private ANN algorithm (NGT (Iwasaki & Miyazaki, 2018)), measured in mean reciprocal ranking (MRR) and recall. PACMANN also has lower latency than linear computation cost algorithms, including Tiptoe (Henzinger et al., 2023) and Preco (Servan-Schreiber et al., 2022) when the database is larger than 5M records in the LAN setting (i.e., when the search engine and the database are co-located, so network round-trip latency is 5 ms), and 50M records in the WAN setting, respectively. For example, for a database with 100M records, Tiptoe requires at least 4s of search latency, whereas PACMANN achieves 1.6s in the LAN setting **(a 60% reduction)** and 3.1s in the WAN setting **(a 22% reduction)**. Experimental details can be found in Section 5.

## 1.1 RELATED WORK

Private Information Retrieval (PIR) (Chor et al., 1995; Chor & Gilboa, 1997; Cachin et al., 1999; Chang, 2004; Gentry & Ramzan, 2005) is a cryptographic primitive that allows a user to retrieve information from a public database without revealing the query to the service provider. Most recent works that rely on Homomorphic Encryption (Melchor et al., 2016; Ahmad et al., 2021; Menon & Wu, 2022; Henzinger et al., 2022) construct communication-efficient PIR schemes with linear computation cost per query. The "client-preprocessing PIR" model (Corrigan-Gibbs & Kogan, 2020; Shi et al., 2021; Corrigan-Gibbs et al., 2022; Zhou et al., 2023; Lazzaretti & Papamanthou, 2023a) introduces a preprocessing phase that allows the client to store sketched information about the database, which can be used to achieve sublinear (amortized) computation and communication cost per query. Most PIR schemes are designed for basic array-type queries. Some PIR schemes consider key-value access (Chor et al., 1997; Patel et al., 2023; Celi & Davidson, 2024) where the client can make private queries based on keyword matching. However, ANN search usually requires a more sophisticated search algorithm and thus requires a more complex access pattern.

Several existing works directly consider the private nearest neighbor search problem. In addition to Tiptoe (Henzinger et al., 2023), Wally (Asi et al., 2024) improves the efficiency of Tiptoe by batching multiple anonymous queries. Nonetheless, Tiptoe and Wally are based on a simple clustering-based searching strategy that partitions the vectors into roughly $\sqrt{n}$ clusters, and only performs exhaustive search in one particular cluster during the search phase. This algorithm is not competitive with the state-of-the-art non-private ANN search algorithms. For example, its top-1 search accuracy is only around 40% of the non-private ANN search algorithms on the MS-MARCO dataset. On the other hand, Preco (Servan-Schreiber et al., 2022) achieves a better search quality with Locality Sensitive Hashing (LSH) techniques, but it requires a linear computation cost per query and a stronger assumption that there are two non-colluding servers storing the same database.

In an independent and concurrent work, Zhu et al.(Zhu et al., 2024) also proposes a privacy-preserving ANN search algorithm under a different setting such that the database is provided by the client and outsourced to the server while being encrypted. In that case, the server not only stores the per-client encrypted database, but can also store per-client state to facilitate the search. Their solution is based on the HNSW graph-based ANN indexing algorithm (Malkov & Yashunin, 2018) and Path Oblivious RAM (ORAM) (Stefanov et al., 2018). Our setting is different that we assume the database is public and shared among all clients, while the server does not have any per-client storage.

## 2 FORMAL DEFINITIONS

$K$**-Approximate Nearest Neighbor Search ($K$-ANN).** Assume a $d$-dimensional metric space with a distance function $\Delta(\cdot, \cdot)$. Given a database containing $n$ vectors, denoted as DB = $\{v_1, v_2, \ldots, v_n\} \in$

$\mathbb{R}^{d \times n}$, and a query vector $q \in \mathbb{R}^d$ such that a $K$-approximate nearest neighbor search algorithm takes the database DB and the query vector $q$ as input, and outputs an index set $I = \{i_1, \ldots, i_K\}$ such that the distances between $q$ and $v_{i_1}, \ldots, v_{i_K}$ are minimized, i.e., $I = \{i_1, \ldots, i_K\}$ is an approximation to the true $K$-nearest neighbors of $q$ in DB. Specifically, we use the recall or the mean reciprocal rank (MRR) to evaluate the quality of the approximation, depending on the context.

**Single Server (Preprocessing) Private $K$-ANN.** A single-server preprocessing private ANN protocol is run between a stateful client and a server. The protocol consists of two phases: preprocessing phase and query phase.

1. **Preprocessing:** The preprocessing phase is run before the query phase and could involve the communication between the client and the server. The server will receive the vector database $\mathsf{DB} \in \mathbb{R}^{d \times n}$ as input.

2. **Queries:** The query phase can include multiple (adaptive) queries from the client. Each query is a vector $q \in \mathbb{R}^d$. The client and the server can have multiple rounds of communication for each query. At the end, the client will output $K$ indices where the corresponding vectors are the $K$-approximate nearest neighbors of $q$ in DB.

The privacy of a private ANN protocol requires that the server learns negligible information about the query vector $q$. Formally, we use a simulation-based definition for the privacy of the protocol, being consistent with prior work (Henzinger et al., 2023).

**Definition 2.1.** An ANN protocol is private if there exists a simulator Sim such that for any probabilistic polynomial-time adversary $\mathcal{A}$ acting as the server, the views of $\mathcal{A}$ in the following two experiments are computationally indistinguishable w.r.t the security parameter $\lambda$:

- Real: the client interacts with $\mathcal{A}(1^\lambda, \mathsf{DB})$ who acts as the server. In each query step, $\mathcal{A}$ may adaptively choose the next query $q$ for the client. The client is invoked with $q$ as input.

- Ideal: the simulated client $\mathsf{Sim}(1^\lambda, n)$ interacts with $\mathcal{A}(1^\lambda, \mathsf{DB})$ who acts as the server and $\mathcal{A}$ still may adaptively choose the next query $q$ for the client. However, the simulator is invoked with only the knowledge of the size of the database, and without the information of the chosen query $q$.

Throughout this paper, we assume the adversary is semi-honest. That is, the adversary follows the server's protocol specification but may try to learn additional information from the server's view. We leave the extension to malicious adversaries as future work.

## 3 OUR GRAPH-BASED ANN CONSTRUCTION

As a starting point, we describe our ANN search algorithm construction without privacy considerations. Our construction relies on a customized version of the graph-based ANN search algorithm.

### 3.1 PRELIMINARY: GENERIC GRAPH-BASED ANN SEARCH BLUEPRINT

Many popular graph-based ANN search algorithms such as NSW (Malkov et al., 2012), HNSW (Malkov & Yashunin, 2018), DiskANN (Jayaram Subramanya et al., 2019), NGT (Iwasaki & Miyazaki, 2018), and FINGER (Chen et al., 2023) follow the same technical blueprint: the algorithm builds a graph based on the input vectors during the preprocessing phase, and then performs a graph traversal algorithm to find the approximate nearest neighbors for a given query vector. We provide a graphical illustration of this process in Figure 3 and a pseudocode description in Figure 4.

**Preprocessing.** The preprocessing phase takes in $n$ vectors $\mathsf{DB} = \{v_1, v_2, \ldots, v_n\} \in \mathbb{R}^{d \times n}$ from a metric space and builds an indexing graph $G$ where the $n$ vertices represent the vectors. Different algorithms may have different ways of selecting graph edges. One common approach (e.g., seen in (Jayaram Subramanya et al., 2019)) is to connect each vertex $v_i$ to several nearest neighbors, i.e. vectors that are close to $v_i$ in the metric space, as well as multiple distant neighbors to ensure the diversity of the neighbor list (Wang et al., 2021). Most recent ANN algorithms use more advanced structures on top of the indexing graph (e.g. hierarchical structure in HNSW (Malkov & Yashunin, 2018) or auxiliary tree-structure in NGT (Iwasaki & Miyazaki, 2018)) to improve the search efficiency.

**Query.** Given a query vector $q$, the query algorithm performs a graph traversal algorithm on the index graph $G$ and outputs a vertex $u^*$ as the approximate nearest neighbor of $q$. The search algorithm starts from an entry point $u_{start}$, often picked as the closest vertex to the centroid of DB. We denote the set of neighbors of a node $v$ as $N(v)$. In each hop, the search algorithm greedily moves from the current vertex $v$ to its neighbor in $N(v)$ that is the closest to the query vector $q$. The algorithm terminates when the number of hops reaches a certain threshold or a local minimum is met, i.e. none of the neighbors is closer to $q$ than the current iterate. This can be easily extended to outputting $K$ approximate nearest neighbors by keeping track of the visited vertices and outputting the top $K$ nearest vertices in the visited set.

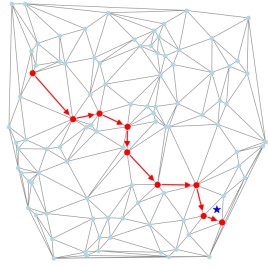

Figure 3: An illustration of the graph-based ANN search algorithm on a 2D space. The starting vertex is located around the upper left part of the graph, and the query vector is shown as a blue star. We highlight the path that the algorithm takes to reach the approximate nearest neighbor.

### 3.2 Our Graph Building Algorithm

To make graph-based ANN search private, PACMANN imposes two constraints: (1) We retain the single graph structure, as opposed to adding auxiliary structures, as is done in the non-private setting. This is done to facilitate the upgrade to the privacy-preserving version. (2) We enforce regularity on the graph's out-degree distribution. This prevents information leakage from the number of neighbors accessed. These properties are not met by existing ANN algorithms, and we customize the graph-building algorithm to meet them. We describe the high-level idea to build a regular $C$-out directed indexing graph here and defer the formal description to Appendix A.

**Start from an unbalanced graph.** Our first observation is that we can take advantage of existing ANN libraries to find nearby neighbors for all vectors in the database and treat them as the candidate neighbors in the graph. Specifically, we find $2C$ approximate nearest neighbors as neighbor candidates for each vertex using an existing ANN library (e.g., NGT (Iwasaki & Miyazaki, 2018)), then trim the candidate list to $C$ candidates with the sparse-neighborhood-graph (SNG) heuristic (Arya & Mount, 1993). Roughly, SNG sorts the candidates by their distance to the vertex and adds candidates to the neighbor list one by one. It only adds a candidate if it is not too close to a neighbor already in the list; this is done to ensure diversity (see Figure 7 for the details). We temporarily add directed edges between each vertex and its $C$ neighbor candidates *in both directions* to ensure the graph is well-connected, i.e. so that each vertex has at least $C$ inbound and $C$ outbound directed edges.

**Balancing the graph.** We now have a graph that could be highly unbalanced in terms of the degrees. We use a sampling technique to balance the graph. That is, for each directed edge $(x \rightarrow y)$, we keep it with a probability of $C/\text{InboundDegree}(y)$. This ensures that a vertex will have $C$ inbound edges in expectation after the sampling process. Then, we ensure that each vertex has exactly $C$ outbound edges. For those vertices with more than $C$ outbound edges, we again use the SNG heuristic to trim the outbounds to $C$. For those vertices with fewer than $C$ outbound edges, we connect them to vertices selected uniformly at random.

## 4 PACMANN: Private Graph-based ANN

We next describe how PACMANN protects query privacy over our customized graph-based ANN search algorithm in Section 3. We start with a basic, inefficient construction, then show how PACMANN optimizes it. Due to space constraints, we present the full construction of PACMANN in Appendix B.

### 4.1 Inefficient Private Graph-based ANN Search with Classical PIR

As a strawman scheme, one could try to protect query privacy by implementing the search algorithm with generic cryptographic primitives, such as fully homomorphic encryption (Gentry, 2009). and/or

Figure 4: Description of the graph based ANN search algorithm including the optional optimizations. The pseudocode can be read in the following two ways: 1) the non-highlighted part describes the generic graph-based ANN search algorithm; 2) the full description, including the optimizations we introduced in Section 4.3, describes our customized version of the algorithm. In the non-private setting, the algorithm is run on the server. In the private setting, the client will run the query algorithm locally, and use Batched Piano PIR to perform the information retrieval dynamically and privately.

---

**Graph-based ANN Search Algorithm**

**Preprocessing.**

- Input: a set of $n$ vectors $\mathsf{DB} = \{v_1, v_2, \ldots, v_n\} \in \mathbb{R}^{d \times n}$.
- Output: a directed graph $G$ with $n$ vertices corresponding to the $n$ vectors in DB, and entry point(s) $u_{\text{start}}, u_{\text{start}}^1, \ldots, u_{\text{start}}^{\sqrt{n}}$.

  1. $G \leftarrow \mathsf{ANN.BuildGraph}(\mathsf{DB})$;      *// Description in Section 3.2*

  2. Let the closest vector to the centroid be the starting vertex $u_{\text{start}} = \arg\min_{u \in [n]} \Delta(v_u, \frac{1}{n} \sum_{j \in [n]} v_j)$.

  3. Let the extra starting vertices be $\sqrt{n}$ random vertices $u_{\text{start}}^1, \ldots, u_{\text{start}}^{\sqrt{n}}$ sampled from $[n]$.

---

**Query.**

- Parameters: $H$: max hop number; beam search width $m$: the number of parallel paths.
- Input: a query vector $q \in \mathbb{R}^d$.
- Output: an index $u^*$ such that $v_{u^*}$ is recognized as an approximate nearest neighbor of $q$.

1. Let $u = u_{\text{start}}$.

2. Let $u_1, \ldots, u_m$ be the top $m$ vertices in $u_{\text{start}}^1, \ldots, u_{\text{start}}^{\sqrt{n}}$ that are closest to $q$.

3. For $t = 1, 2, \ldots, H$:

   - Update $u$ as follows:
     – Read the neighbor list $N(u)$ from $G$ and all vectors $v_j$ for $j \in N(u)$ from DB.
     – Let $u' = \arg\min_{j \in N(u)} \Delta(v_j, q)$.
     – If $\Delta(u', q) \leq \Delta(u, q)$, update $u \leftarrow u'$.
   - Similarly, update $u_1, \ldots, u_m$.

4. Let $u = \arg\min_{u' \in \{u, u_1, \ldots, u_{\sqrt{n}}\}} \Delta(u', q)$.

5. Output $u^* = u$.

---

multi-party computation (Cramer et al., 2015). However, the search algorithm of graph-based ANN is inherently iterative and adaptive in that during the graph traversal process, each hop's vertex is dynamically determined by both the previous hop's vertex and also the query vector. Handling such (adaptive) iteration is a common challenge in designing cryptographic protocols [2] as observed by previous works (Barni et al., 2009; Delpech de Saint Guilhem et al., 2022; Choudhuri et al., 2024; Goel et al., 2022; Heath et al., 2021)

**Localized Searching.** Our main idea is to *perform the iterative graph traversal algorithm on the client side*, completely removing the privacy concern of performing iterative computation on the server. Naively, this would require the client to store the whole indexing graph, which is impractical in terms of storage cost. Instead, we let the client dynamically and privately retrieve subgraph information from the server. Specifically, during search, the client can retrieve the neighbor list $N(v_i)$

---

[2] The standard FHE or MPC techniques are designed for the so-called circuit model, where the computation flow is fixed and known in advance. Although the circuit model is indeed Turing-complete, it does not naturally support a random-access memory and control flow operations like if-else conditions. In the context of graph-based ANN search, we do need to (adaptively) read the graph from a random access memory, so the standard techniques are not directly applicable.

of the current vertex $v_i$ and also all vectors in the neighbor list from the server for each hop. The client then computes the distances between the query vector $q$ and all the neighbor vectors in $N(v_i)$ locally, and chooses the closest neighbor within that set. This process is repeated until the search terminates.

**Protecting Retrieval Privacy with PIR.** Even without directly observing the query vector, the server can infer non-trivial information from the vertex indices retrieved by the client. For example, if the server learns that the client retrieves many vertices whose underlying records represent documents about food topics, the server can infer that the client is searching for information about foods. Thus, the final challenge is how the client can retrieve the graph information (including the neighbor lists and the vectors) without revealing which vertex it is retrieving.

We use a Private Information Retrieval (PIR) scheme for this purpose. Standard PIR (Chor et al., 1995) allows a client to retrieve one or multiple entries from a server-stored database of $n$ entries without revealing the indices of the entries to the server.

To integrate a PIR scheme into our private graph-based ANN search algorithm, we can run a standard PIR protocol for a database storing all the vertex information in the graph, where each entry $i$ stores the vector $v_i$ and also the neighbor list $N(i)$. Whenever the client needs to access the graph, it issues a PIR query to retrieve the information privately. Notice that by merging the vector information and also the neighbor information into a single database, we can finish each hop in one round of communication.

**Efficiency Issue.** This construction runs into a major efficiency issue. It was formally proved by Beimel, Ishai, and Malkin (Beimel et al., 2000) that any PIR scheme requires $\Omega(n)$ computation cost if there is no preprocessing, and if the server stores the database naively without encoding. Hence, if we have a non-private graph-based ANN algorithm that proceeds for $H$ hops, and each hop requires the client to retrieve the information of $C$ neighbors, the online computation cost of the scheme will be $\Omega(HCn)$, which is super-linear in $n$. This is worse than the prior linear-cost private ANN schemes (Henzinger et al., 2023; Servan-Schreiber et al., 2022).

### 4.2 Improving Efficiency with Preprocessing PIR

To address this efficiency limitation, we utilize a new paradigm of PIR schemes: client-side preprocessing PIR (Corrigan-Gibbs & Kogan, 2020). Client-side preprocessing PIR achieves sublinear (amortized) query cost by allowing a one-time preprocessing phase before the query phase in which the client will eventually store useful information about the database. Specifically, we use a PIR scheme called Piano (Zhou et al., 2024) that achieves practical efficiency for the single-server setting, matching our requirements. Given a database with $n$ entries, Piano achieves $\tilde{O}(\sqrt{n})$ computation and communication per query with $\tilde{O}(\sqrt{n})$ client storage after a one-time preprocessing phase that incurs linear computation and communication costs.[3] Other alternatives of PIRs (Ghoshal et al., 2024; Ren et al., 2024) could also be used in our construction for similar efficiency guarantees.

By applying the Piano PIR scheme to our private ANN construction, after a linear cost preprocessing phase, each PIR query in the graph traversal algorithm will incur $\widetilde{O}(\sqrt{n})$ computation and communication cost. Again, if we assume the search algorithm takes $H$ hops and each hop requires the client to retrieve the information of $C$ neighbors, the overall running time of the scheme will be $\widetilde{O}(HC\sqrt{n})$. As long as $H \cdot C$ is $o(\sqrt{n})$, the resulting scheme will have sublinear computation cost, being more efficient than the prior private ANN schemes (Henzinger et al., 2023; Servan-Schreiber et al., 2022).

**Tradeoff and Practicality of Preprocessing.** Despite significant online efficiency improvements, the Piano PIR client must download the whole database (i.e., the indexing graph) in a streaming fashion during the preprocessing phase. Here, streaming means that the client only stores a small portion of the database to preprocess at a time, but the total amount of data the client has to download is still the whole database. Therefore, our scheme is more suitable for a client with a good network connection. In our evaluation where we assume a good network connection (1 Gbps), the preprocessing phase takes around 5 minutes and 3 GB of client storage space for a database with 100 million vectors, and the total download communication is around 60GB. An alternative approach to reduce the

---

[3]We use the notation $\widetilde{O}(\cdot)$ to hide polylogarithmic factors.

preprocessing cost is to have a second server to preprocess the database and directly provide the client with the preprocessed results, as suggested in many existing PIR schemes (Zhou et al., 2024; Lazzaretti & Papamanthou, 2023b; Ghoshal et al., 2024; Kogan & Corrigan-Gibbs, 2021).

### 4.3 NECESSARY OPTIMIZATIONS

In practice, we observe that the total number of visited vertices in the graph-based search algorithm (that is, the product between max hop number $H$ and the graph out-degree $C$) tends to be around a few hundreds or thousands. We describe the necessary optimizations here and conduct an ablation study in the evaluation section to show the effectiveness of these optimizations in Section 5. Empirically, we observe that our optimizations reduce the concrete computation cost by 76%, and the overall latency by nearly 70% (including the communication time).

**Beam Search.** The basic description of the graph-based ANN search algorithm Figure 4 traverses the graph in a single path. In practice, we can explore m multiple paths in parallel. This approach is used in other graph-based ANN algorithms (Jayaram Subramanya et al., 2019). Although doing so increases the query cost per hop, we observe that beam search significantly reduces the total hops to reach a given search quality, and is thus essential to reducing the overall latency.

**Fast Starting.** Intuitively, if the starting vertex is already very close to the query vector $q$, we can reach the approximate nearest neighbors with fewer hops. With fast starting, we let the client store around $O(\sqrt{n})$ vertices' information locally, and scan all these vertices to find those that are already close to $q$ to start the searching algorithm. We get this benefit for free, because Piano PIR already requires the client to store $\tilde{O}(\sqrt{n})$. vectors locally, which are selected uniformly at random.

**Batched PIR Query.** We observe that the PIR queries in each hop are parallel and can be handled in a single batch. We can use a batching technique called partial batch retrieval (PBR) (Servan-Schreiber et al., 2022) to further improve the efficiency. We provide the details in Appendix B.2.

## 5 EVALUATION

We evaluate PACMANN's performance in terms of search quality and latency and compare it against two baselines: a state-of-the-art private ANN search algorithm, Tiptoe (Henzinger et al., 2023), and a non-private ANN search algorithm, NGT (Iwasaki & Miyazaki, 2018). Our evaluation results show that PACMANN achieves better search quality with lower online query latency than Tiptoe in two real datasets, SIFT and MS-MARCO. We provide more details in Appendix C, including a detailed breakdown of the preprocessing, storage, and communication cost, and a discussion about quantization and the alternative implementations.

### 5.1 EVALUATION SETUP

Our open-source implementation[4] uses Python for data preprocessing and Golang for the core algorithm, including the graph ANN algorithm and the PIR scheme. Details of the implementation are provided in Appendix C. The experiments are run on a single server with a 2.4GHz Intel Xeon E5-2680 CPU and 256 GB of RAM. We evaluate the latency numbers on two simulated network settings, the local area network (LAN) setting with 5ms round-trip-time (RTT) and the wide area network (WAN) setting with 50ms RTT. Although our implementation supports multi-threading optimization, we only use a single thread for all the experiments for a fair comparison.

**Quality Metrics**

*Recall:* Recall is the standard metric used in evaluating the quality of ANN algorithms (Aumüller et al., 2020). For each query $q$, if the algorithm outputs a $K$-index set $I$, and the top $K$ ground truth indices are $I^*$, the recall@K is defined as: $\text{Recall@}K = \frac{|I^* \cap I|}{K}$.

*Mean Reciprocal Rank (MRR):* MRR is a standard quality metric for information retrieval systems (Nguyen et al., 2016). Given a query $q$, assume there is a ground truth index $i^*$ such that $\text{DB}[i^*]$

---

[4] https://github.com/wuwuz/Pacmann

is the most relevant entry. If the client outputs a list of indices that actually contains $i^*$ at the $j$-th rank where $j \leq K$, the reciprocal rank (denoted as RR@$K$) score is $1/j$. Otherwise, RR@$K$ is 0. MRR is defined as the average of RR@$K$ over multiple queries.

## Datasets

*SIFT Dataset (Jégou et al., 2011)* We use the first 100 million 128-dimensional vectors from the SIFT dataset for our evaluation. We vary the database size by picking the first 2 million to 100 million vectors. We test 1,000 top-10 queries in the query set for each configuration, and measure the average recall@10 (the top-10 ground truth nearest neighbors for each query are provided by the dataset).

*MS-MARCO Dataset. (Nguyen et al., 2016)* The dataset contains 3.2 million text documents with more than 5000 test queries such that the top 1 relevant document is provided. We follow the same procedure as in the Tiptoe paper (Henzinger et al., 2023) to process the dataset: 1) embed the documents and the queries into a 768-dimensional vector space with sentence-BERT (Reimers & Gurevych, 2019); 2) reduce the dimensions to 192 with PCA. We follow the same quality evaluation metric as in the Tiptoe paper that we compute the average MRR@100 for the first 1000 queries.

## Baselines

*Tiptoe (Simulated).* We aim to compare our scheme against the state-of-the-art private ANN search algorithm, Tiptoe (Henzinger et al., 2023). Due to resource constraints, we were unable to run the full Tiptoe system, in which the dominating cost comes from the clustering step that requires dozens of servers running hundreds of core-hours as reported in the original paper. Hence, we simulated Tiptoe by using two baselines as follows.

- Latency lower bound: *Linear Algorithm.* The Tiptoe algorithm computes $n$ inner products per query with preprocessed homomorphic encryption. As a latency lower bound, [5] we implement a linear time algorithm that only computes the inner product between the query and each database vector in plaintext. We optimized the implementation using AVX-512 instructions. The throughput of this algorithm is approximately 11.9 GB/s, matching the memory bandwidth of the machine. Finally, we do not include network latency for this baseline.
- Quality upper bound: *Cluster.* We replicate the clustering-based ranking algorithm used in Tiptoe with full precision (32-bit)[6]. We cluster the vectors into $\sqrt{n}$ clusters offline. For each query, we find the closest cluster centroid, and search for the nearest neighbor within the cluster.

*NGT.* NGT is one of the state-of-the-art non-private ANN search algorithms. We use the GoNGT library (Iwasaki & Miyazaki, 2018). We compare our scheme against NGT in terms of search quality.

## 5.2 EVALUATION RESULTS

**PACMANN provides a favorable tradeoff between privacy, search quality, and latency.** We first measure the tradeoff between search quality and query latency. Specifically, by increasing the number of rounds and the number of parallel queries in each round, our algorithm can achieve higher search qualities at the cost of higher latency. The results are shown in Figure 2. We consider a wide range of search quality requirements, where the lower bound is set by the cluster search algorithm (replicating the Tiptoe search algorithm), and the upper bound is set by the non-private ANN algorithm, NGT. We observe that our scheme provides a wide range of tradeoffs between quality and latency, and can indeed achieve approximately 90% of NGT's search quality. In the SIFT-100M dataset, the advantage of our scheme is more significant, and even with 91% recall@10, our scheme still has a lower latency than the linear algorithm baseline.

**Scalability: PACMANN outperforms Tiptoe in latency and accuracy on datasets of at least 2M-50M records.** We next evaluate the scalability of our scheme by scaling the database size up to 100 million vectors. We tune the parameters of our scheme to achieve 0.90 recall@10 for each

---

[5]We compared the simulated online latency with the actual numbers reported in the original paper (Henzinger et al., 2023), and the simulated latency is 8% less than the reported latency. See the detailed results in Appendix C.

[6]Our reimplementation of the clustering-based algorithm reached 0.15 MRR@100 on the MS-MARCO dataset, better than the 0.13 MRR@100 reported in the Tiptoe paper (Henzinger et al., 2023).

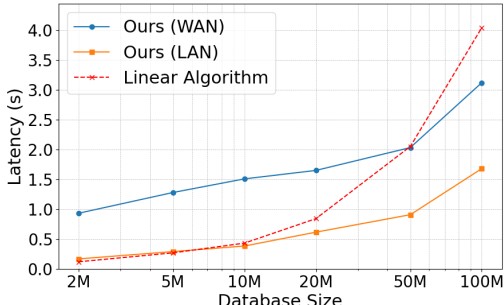 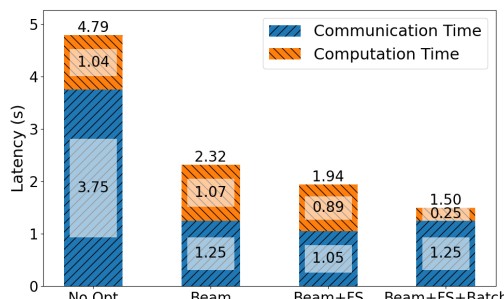

Figure 5: Latency results on different database sizes, sampled from the SIFT dataset. We tune the parameters of our scheme to achieve 0.90 recall@10 for each data point; For each data point, the total latency is the sum of the actual computation time and the round-trip time of the communication, multiplied by the number of rounds. We plot both the LAN setting (5ms rtt) and the WAN setting results (50ms rtt) in this figure.

Figure 6: Ablation study on the 10M subset of SIFT (WAN setting). Given a fixed configuration, we increase the max number of rounds until the quality reaches 0.90 recall@10. "No Opt" means no optimizations are enabled. "Beam" means the beam search optimization. "FS" means the fast starting strategy. "Batch" means we enable the batch-mode Piano PIR. Our full implementation enables all the optimizations.

data point; that is, 9 out of 10 of the search results are indeed ground truth top-10 nearest neighbors on average. The results are shown in Figure 5. We observe that in the LAN setting where the computation time is dominant, our scheme beats the linear algorithm baseline when the database size is larger than 5M. In the WAN setting where the communication time is more dominant, our scheme beats the linear baseline when the database size is larger than 50M. We observe that the number of rounds to achieve a certain recall@10 increases roughly logarithmically with the database size, and we know from the theoretical analysis that PIR cost scales with $\sqrt{n}$. This suggests that the total latency of our scheme increases sublinearly with the database size.

**Ablation study: Our optimizations give a 70% latency reduction.** Finally, we conduct an ablation study on the 10M subset of the SIFT dataset to evaluate the optimizations in Section 4.3: 1) *Beam Search:* The number of parallel paths in the beam search algorithm is increased from 1 to 3. 2) *Fast Starting:* The client starts the search by choosing the starting vertices from $\widetilde{O}(\sqrt{n})$ preprocessed vertices. 3) *Batched PIR:* We enable the batch-mode PIR for each round of the search.

We compare four configurations in Figure 6, where we increase the number of rounds until the quality reaches 0.90 recall@10. The beam search optimization significantly reduces the maximum number of rounds needed to achieve the same quality by 3x. Adding the fast-start strategy furtuher reduces the required rounds by 20%. When we enable the batch-mode PIR, we observe that the computation time in each round is reduced by 4x, but, as we mentioned in Appendix B.1, we do introduce some query failures, which we need to balance with the slight increase in the number of rounds.

## 6 CONCLUSION

We present PACMANN, a new private ANN search scheme that allows clients to perform nearest neighbor search queries over hundreds of millions of vectors while preserving the queries' privacy. PACMANN achieves significantly better search quality compared to the state-of-the-art private ANN search schemes and has lower latency in large-scale datasets. PACMANN could be applied to a wide range of applications including conventional search and retrieval-augmented generation.

**Limitations.** PACMANN inherits the drawbacks of the single-server preprocessing PIR schemes and requires the client to download the whole indexing structure offline in a streaming manner. Therefore, PACMANN will not be suitable for network-constrained scenarios. PACMANN also does not naturally support dynamic updates to the database, which has been a challenging open problem in the field of PIR. Finally, PACMANN is designed under the assumption that the database is public, so we do not consider server-side privacy. We leave these questions as interesting future directions.

## REPRODUCIBILITY STATEMENT

Our open source implementation can be found at the anonymous repository on Github: `https://github.com/wuwuz/pacmann`. The repository contains the following components:

- Instructions to install the required dependencies;
- Our own implementation for PACMANN;
- All baseline implementations including NGT, clustering-based ANN and also the plaintext inner-product search (referred as "Linear algorithm" in Section 5);
- A script to download the SIFT dataset;
- Customizable scripts to run the baselines and our implementation with different parameter settings.

## ACKNOWLEDGMENT

This work is in part supported by a grant from ONR, a grant from the DARPA SIEVE program under a subcontract from SRI, a gift from Cisco, Samsung MSL, NSF awards under grant numbers 1705007, 2128519 and 2044679. In addition, this work is in part supported by NSF grant CCF-2338772, as well as support from the Sloan Foundation, Intel, and Bosch. We thank Justin Zhang for his contribution in the evaluations.

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

# A   DETAILS OF OUR GRAPH-BASED ANN CONSTRUCTION

We now provide a detailed description of our customized graph building algorithm in Figure 7.

---

**Our Graph Building Algorithm: ANN.BuildGraph.**

**Input:** a set of $n$ vectors $\mathsf{DB} = \{v_1, v_2, \ldots, v_n\} \in \mathbb{R}^{d \times n}$ and a target degree $C$.

**Output:** a directed regular $C$-out graph $G$ with $n$ vertices corresponding to the $n$ vectors in DB.

---

**Subroutine:** ANN.TrimNeighbors$(u, L, C)$. *// Given a vertex $u$ and a list of neighbors $L$, return the picked $C$ neighbors of $u$ in $L$ with the SNG heuristic.*

1. Let $w$ be the number of vertices in $L$ and let $(i_1, \ldots, i_w)$ be the sorted indices in $L$ according to the distance to the vector $v_u$ (ascendingly).

2. Let $\mathsf{Kept} \leftarrow \emptyset$, $\mathsf{Dscd} \leftarrow \emptyset$.

3. For $j = i_1, \ldots, i_w$:

    (a) If $\exists j' \in \mathsf{Kept}$ such that $\Delta(v_j, v_{j'}) < \Delta(v_j, v_u)$, label $j$ as discarded. *// SNG heuristic*

    (b) If $j$ is not discarded, insert $j$ to $\mathsf{Kept}$. Otherwise, insert $j$ to $\mathsf{Dscd}$.

    (c) Terminate if $|\mathsf{Kept}| = C$.

4. If $|\mathsf{Kept}| < C$, insert the first $C - |\mathsf{Kept}|$ vertices in $\mathsf{Dscd}$ to $\mathsf{Kept}$.

5. Return $\mathsf{Kept}$.

---

**Graph Building.**

1. For each $u \in [n]$:

    (a) Let $N_u$ be the set of $2C$ approximate neighbor indices to $v_u$ obtained by an underlying ANN algorithm.

    (b) Trim $N_u$ to $C$ neighbors: let $N_u \leftarrow$ ANN.TrimNeighbor$(u, N_u, C)$.

    (c) For all $u' \in N(u)$, add directed edges $(u \rightarrow u')$ and edge $(u' \rightarrow u)$ to the graph $G$.

2. Fix $\mathsf{InDegree}(u)$ to be the in-degree of vertex $u$ in the current graph $G$.

3. For each edge $(i, j) \in G$: keep the edge in $G$ with probability $\frac{C}{\mathsf{InDegree}(j)}$.

4. For each $u \in [n]$:

    (a) Denote $N_G(u)$ be the set of outbounds of $u$ in $G$.

    (b) If $|N_G(u)| < C$: add edges from $u$ to random vertices in $[n]$ until $|N_G(u)| = C$.

    (c) If $|N_G(u)| > C$: let $N_G(u) \leftarrow$ ANN.TrimNeighbor$(u, N_G(u), C)$.

5. Output the graph $G$.

Figure 7: Description of our graph building algorithm ANN.BuildGraph.

# B   DETAILED CONSTRUCTION DESCRIPTION

We include the full description of our algorithm in Figure 8. The algorithm further involves the local caching optimization that the client will keep track of all queried vertices and avoid repetitive queries.

## B.1   REVIEW ON PIANO PIR

We review the main idea of Pian PIR (Zhou et al., 2024) below. During the preprocessing phase, the client will sample around $\sqrt{n} \log n$ random sets, each containing $\sqrt{n}$ random indices in $[n]$. For each sampled set $S$, the client will store the random seed to generate $S$, and also the sum of the entires indexed by $S$, denoted as $\mathsf{sum}(S) = \sum_{i \in S} \mathsf{DB}[i]$. The number of sets ensure the every entry in DB is included in at least one set. Then, for each query $x$, the client will first find a local set $S$ that contains $x$. Since the client knows $\mathsf{sum}(S)$, if it can learn learn $\mathsf{sum}(S/\{x\})$, it can

recover $DB[x]$ by a simple subtraction. However, the client cannot query $sum(S/\{x\})$ by directly sending those $\sqrt{n} - 1$ indices to the server, because the server will notice that all the indices it received from the client are not $x$, which is a privacy issue. Piano fixes this issue by having the client remember roughly $\tilde{O}(\sqrt{n})$ random entries during the preprocessing phase, which are called the "replcaments". Then, instead of deleting $x$ from $S$, the client will replace $x$ with a replacement entry $y$ and query the server of $sum(S/\{x\} \cup \{y\})$ by sending all $\sqrt{n}$ indices in this "edited" set to the server. The server takes $\sqrt{n}$ computation to compute the sum and returns the result to the client. Since the client already knows the value of $DB[y]$ in preprocessing, it can recover $DB[x]$ by calculating $DB[x] = sum(S) - sum(S/\{x\} \cup \{y\}) + DB[y]$. It is not hard to see this single-query version of Piano is indeed private – the server only sees $\sqrt{n}$ random indices that are independent of the actual query index $x$. We omit the details of the multiple, adaptive-query version of Piano here and refer the interested readers to the original paper (Zhou et al., 2024).

**Batched Piano PIR.** We can use a technique called Partial Batched Retrieval to support multiple queries in a single round of communication. The idea is to use a pseudorandom permutation to permute the database upfront, and then partition the permuted database into $B$ sub-databases where each sub-database has $n/B$ entries. On average, each sub-database will have $Q/B$ queries given a batch of $Q$ queries (if we are using beam search, $Q = mC$). By doing this, each PIR query will be issued to a sub-database of size $n/B$ instead of the full database, saving the computation and communication cost. To ensure privacy, the client has to issue a fixed number of queries for each sub-database, say choosing the number $T = 1.5Q/B$. If fewer than $T$ queries are in the same sub-database, the client submits dummy queries. If more than $T$ queries are in the same sub-database, the client only submits the first $T$ queries and discards the rest. For example, if we have a batch size $Q = 32$ and set partition number $B = 16$ and $T = 3$, around 90% of the queries in the batch will be successfully submitted in expectation. [7] Intuitively, if each vertex has 32 neighbors, our retrieval process can retrieve nearly 30 neighbors' information with a single batched query. Since the graph tends to be highly-connected, we observe that even with a mild query failure probability, the algorithm still finds good results with more hops.

In the algorithm description, we use the following syntax of to capture the Piano PIR functionality with the batch-mode interface:

- hint $\leftarrow$ PIR.Prep($1^\lambda$, DB): given the security parameter $1^\lambda$ and the database DB, generate the preprocessed PIR hint hint.
- msg $\leftarrow$ PIR.BatchQuery(hint, batch): given the PIR hint and a batch of query indices batch $\subseteq [n]$, the client generates the PIR query message msg.
- ans $\leftarrow$ PIR.BatchAnswer(DB, msg): upon receiving the batched PIR query message msg, the server generates the batched answer ans.
- $(\beta, \text{hint}') \leftarrow$ PIR.BatchRecover(hint, ans, batch): given the batched answer ans and the PIR hint hint, recover the answers $\beta$ and update the hint to hint'. Here, $\beta$ is a set that includes all the successful query indices and their corresponding answers. That is, $\beta$ includes the tuples $(i, DB[i])$ for all $i \in$ batch that are successfully queried.

## B.2 FULL ALGORITHM DESCRIPTION

**Preprocessing.** We use the same graph building algorithm as in Figure 7 to build a graph $G$ from the vectors DB. We then combine the neighbor list of each vertex with its vector to build a new database $DB' = \{(v_i, N(i))\}_{i \in [n]}$. Then, we randomly sample $\sqrt{n}$ starting vertices $u_{\text{start}}^1, \ldots, u_{\text{start}}^{\sqrt{n}}$ from $[n]$. We then run the client-side preprocessing algorithm of Piano and let the client store the hint hint. Further, we let the client download those starting vertices and their vectors and neighbor lists from the server.

---

[7] We can approximate this process with randomly throwing $Q$ balls into $B$ bins, each with a maximum load of $T$, and calculate how many balls are discarded due to overflow in expectation. Given some fixed order of throwing the balls, the probability that the $i$-th thrown ball is thrown into a fully-loaded bin is always $\Pr[\text{Bin}(i - 1, 1/B) \geq T]$. Here, $\text{Bin}(t, p)$ denotes the Binomial distribution – that is, the number of successful trials in $t$ repeated, independent trials where each trial has success probability of $p$. Then, the expected number of discarded balls (i.e., failed queries) in total is $\sum_{i=1}^{Q} \Pr[\text{Bin}(i - 1, 1/B) \geq T]$.

**Query for $K$-approximate nearest neighbors of vector** $q \in \mathbb{R}^d$. We generalize the search algorithm to the following exploration process. The client can maintain two sets of vertices locally: 1) visited including all vertices that the client has already visited and stored their vectors and neighbor lists, and 2) developed including all vertices that the client has already visited and retrieved all their neighbors' information. The client can maintain visited as a priority queue where the closer vertices to $q$ are at the front. Each time the client can pick multiple vertices from visited and fetch their neighbors' information in parallel. The client will move those picked vertices from visited to developed, and add their neighbors to visited. We collect all those neighbors' indices as batch and issue a batched PIR query to the server. After receiving the batched answer, the client will recover all the successful queries and push them to visited. The client also needs to update the local hint hint after each query. We can repeat this process until the max hop number is reached. Finally, the client can output the top $K$ nearest vertices in visited and developed to the query vector $q$.

We summarize the main result in the following theorem.

**Theorem B.1.** *Assume there exists one-way functions. Consider a database with $n$ vectors in $\mathbb{R}^d$. Assuming a non-private graph-based ANN algorithm that preprocesses the database into a $C$-degree graph, and searches for the approximate nearest neighbors with $H$ hops in the graph. Then, by applying the Piano PIR scheme, we can build a private ANN scheme with the following efficiency, while achieving the same search quality as the non-private graph ANN:*

- *Preprocessing: $\tilde{O}(n(d+C))$ client and server time and $O(n(d+C))$ communication;*

- *Query: $\tilde{O}(H \cdot \sqrt{Cn} \cdot (d+C))$ client and server time and $O(H \cdot \sqrt{Cn} \cdot (d+C))$ communication.*

- *Storage: $\tilde{O}(\sqrt{n} \cdot (d+C))$ client storage and no additional server storage except the database.*

**Proof.** The proof follows directly from the efficiency of the Piano PIR scheme when we consider a database with $n$ entries and each entry stores a vector in $\mathbb{R}^d$ and a neighbor list of size $C$. Moreover, by applying the batched optimization we described in Section 4.3, we further reduce the computation and communication cost of each PIR query by $\sqrt{C}$. The privacy proof is fairly straightforward, as the server only sees multiple Piano PIR queries from the client. We can construct a simulator for the Ideal experiment that simulates the server's view by invoking the simulator of the Piano PIR scheme for each query issued from the client. It follows from the privacy of the Piano PIR scheme that the two views (the Real and the Ideal experiments) are computationally indistinguishable. ∎

## C    DETAILED EVALUATION

### C.1    IMPLEMENTATION DETAILS

We describe our implementation of the two components below.

**Graph-based ANN.** The graph-based ANN algorithm can be divided into two parts: the graph building part and the query part. The graph building part follows our description in Figure 7 and sets the outbound number $C = 32$. The query part strictly follows our description in Figure 8.

**Batched PIR.** We implement a batch-mode version of the Piano PIR scheme (Zhou et al., 2024) by first implementing the single-query version of the Piano PIR scheme with 128-bit security parameter, then wrapping it with a batch-mode interface. The interface simply splits the whole database into $B$ partitions, and then runs a single-query Piano PIR scheme on each partition parallelly. Given a batch of $Q$ queries, we simply identify which partition each query belongs to, and then make $Q/B$ queries at each partition to ensure privacy. We choose $B = 16$ and $Q = 32$ in our evaluation. Moreover, we implement the automatic maintenance for the client state. We keep track of the consumed hints in the client's space and automatically run another preprocessing phase when the hints are exhausted.

### C.2    DETAILED BREAKDOWN

Finally, we provide a detailed breakdown of the costs of our scheme in Table 1. We pick two representative parameter settings for the MS-MARCO dataset and the SIFT dataset, all achieving

---

**PACMANN: Private Graph ANN with Preprocessing**

**Preprocessing.**

- Server Side Preprocessing:         *Input: a set of $n$ vectors* $\mathsf{DB} = \{v_1, v_2, \ldots, v_n\} \in \mathbb{R}^{d \times n}$.
    - Runs $G \leftarrow \mathsf{BuildGraph}(\mathsf{DB})$.
    - Randomly sample $\sqrt{n}$ starting vertices $u_{\text{start}}^1, \ldots, u_{\text{start}}^{\sqrt{n}}$ from $[n]$.
    - Let $\mathsf{DB}' = \{(v_i, N(i))\}_{i \in [n]}$ where $N(i)$ is the neighbor list of $i$ in $G$.

  - Client Side Preprocessing:
      - Stores hint $\leftarrow \mathsf{PIR.Prep}(1^\lambda, \mathsf{DB}')$.     *// Involving communication with the server.*
      - Downloads the representative indices $u_{\text{start}}^1, \ldots, u_{\text{start}}^{\sqrt{n}}$ from the server.
      - For all $u \in \{u_{\text{start}}^i\}_{i \in [\sqrt{n}]}$, downloads $\mathsf{DB}'[u] = (v_u, N(u))$ from the server.

---

**Query for $K$-approximate nearest neighbors of vector $q \in \mathbb{R}^d$.**

  - Parameters: max hop number $H$ and beam search width $m$.

- Client Side Algorithm.
    - Let visited $= \{u_{\text{start}}^i\}_{i \in [\sqrt{n}]}$ be a priority queue where the closest vertices to $q$ are at the front.
    - Let developed $= \emptyset$.
    - For $t = 1, 2, \ldots, H$:
        * Let $u_1, \ldots, u_m$ be the top $m$ indices in visited and move them from visited to developed.
        * Let batch $= \cup_{i \in [m]} N(u_i)$ be the union of the neighbor lists of $u_1, \ldots, u_m$.
        * Remove those indices in batch that are already in visited $\cup$ developed.
        * Send msg $\leftarrow \mathsf{PIR.BatchQuery}(\text{hint}, \text{batch})$ to the server.
        * Upon receiving ans from the server, run $(\beta, \text{hint}') \leftarrow \mathsf{PIR.Recover}(\text{ans}, \text{hint})$. where $\beta$ is a set that includes all the successful query indices and their corresponding answers.
        * For each successful query tuple $(u, (v_u, N(u))) \in \beta$, add $u$ to visited and store the vector $v_u$ and neighbor list $N(u)$ locally.
        * Update the local PIR hint hint $\leftarrow \text{hint}'$.
    - Finally, for all $u \in$ visited $\cup$ developed, compute the distance $\Delta(v_u, q)$. Output the $k$ indices with the smallest distances.

- Server Side Algorithm.
    - Upon receiving a PIR batch query message msg from the client, run ans $\leftarrow \mathsf{PIR.BatchAnswer}(\mathsf{DB}', \text{msg})$ and return ans.

---

Figure 8: Our private ANN scheme with preprocessing.

|  | MS-MARCO (3M) | SIFT (100M) |
|---|---|---|
| **Preprocessing** | | |
| Graph Buliding Time | 8.5 min | 343.5 min |
| PIR Preprocessing | 9.1 s | 271.6 s |
| Communication | 2.7 GB | 59.6 GB |
| **Online per query** | | |
| Latency | 1.1 s | 3.0 s |
| Computation Time | 0.10 s | 1.48 s |
| Online Communication | 1.5 MB | 14.4 MB |
| Rounds | 20 | 32 |
| **Maintenance per query** | | |
| Time | 0.19 s | 1.99 s |
| Communication | 60.1 MB | 399.4 MB |
| Client Storage | 0.6 GB | 2.9 GB |

Table 1: Detailed breakdown of our results on different datasets in the WAN setting. We list the preprocessing, online query, and maintenance costs. The graph-building cost happens only once. The PIR preprocessing cost is incurred when each client joins the system. The online query cost is the cost on the critical path of the search queries. Following each online query, there is a necessary one-round maintenance to update the client state. We use 16 threads for the graph building and one thread for other parts. The corresponding quality for the experiments is 0.266 MRR@100 for MS-MARCO and 0.90 recall@10 for SIFT.

90% of the quality of the state-of-the-art non-private ANN search algorithm. The graph building time is significant for both datasets, taking 8.5 minutes for the MS-MARCO dataset and 343.5 minutes for the SIFT dataset. However, the graph building only happens once on the server side. The PIR preprocessing is per-client, but it is relatively cheap, taking 9.1 seconds for the MS-MARCO dataset and 271.6 seconds for the SIFT dataset. The most expensive part is that during the preprocessing, the client has to scan over the whole index structure in a streaming fashion, incurring large communication cost. For each online query, the latency and computation time match our analysis in Section 5.2. Notably, the communication cost on the critical path is relatively low, taking 1.5MB for the MS-MARCO dataset and 14.4MB for the SIFT dataset. Notice that the client has to update its local state after each query. We see that the maintenance time per query is relatively cheap, but the communication cost is high, taking 60.1MB for the MS-MARCO dataset and 399.4MB for the SIFT dataset. Theoretically, the client stores around $\widetilde{O}(n)$ amount of local state. Empirically, we see that the client stores 0.6GB of data for the MS-MARCO dataset and 2.9GB of data for the SIFT dataset, and the scaling factor matches our theoretical analysis.

**Discussion on quantization and comparing against Tiptoe.** We noticed that the original Tiptoe implementation uses 4-bit quantization for the vectors to further trade off the search quality for the latency. In our evaluation, we use the full 32-bit precision for all the algorithms. Thus, the quality of the clustering-based algorithm should be considered an upper bound for the actual Tiptoe algorithm. For the latency, the "Linear Algorithm" baseline should be a good approximation of the actual Tiptoe algorithm. As an example, if we extrapolate the reported numbers of Tiptoe's algorithm based on the Table 6 and Table 7 in the original paper (Henzinger et al., 2023) to the MS-MARCO dataset, the latency will be around 0.27 seconds. [8]

**Alternative Implementations.** As alternatives, one can indeed use other PIR schemes to implement our graph-based ANN algorithm. As an example, using another recent single-server PIR scheme,

---

[8]The authors of Tiptoe reported the total core-second is 145 seconds for their experiment on a dataset with 360 million 192 dimensional vectors. The full Tiptoe system contains three parts: "preprocessing", "ranking" and "URL" where the "ranking" part corresponds to our ANN search experiment. According to Table 7 in the paper, it should take about $(1.9/(6.5 + 1.9 + 0.6)) = 21\%$ of the cost. Thus, an estimation of the time on a single-core for the 3.2M size MS-MARCO dataset will be $145 \times 21\% \times \frac{3.2 \times 10^6}{360 \times 10^6} \approx 0.27$s, which is close to our simulated "Linear Algorithm" latency (0.25 seconds).

SimplePIR (Henzinger et al., 2022) (used also in Tiptoe (Henzinger et al., 2023)), comes with a tradeoff. In the 10M SIFT dataset experiment, we can save the offline communication cost from roughly 6GB to around 300MB, but this would increase the online latency from 1.5s to roughly 90s according to our estimation as follows. We tested the throughput of the SimplePIR scheme on our server, which is around 10 GB/s. Then, we are making in total 25 rounds of graph explorations where each round having 96 parallel queries. The batching technique allows each individual query to be made in a sub-database of 16 times smaller than the whole database (around 6GB). Therefore, the total estimated time would be $25 \times 96 \times \frac{6\text{GB}}{16 \times 10\text{GB/s}} \approx 90\text{s}$.

## D    THEORETICAL IMPLICATIONS

We focused on the practical aspects of our scheme in the main body. Although graph-based ANN search has been empirically shown to be successful in practice, the theoretical understanding of the graph-based ANN search is still very limited. Here, we provide some theoretical insights into the graph-based private ANN search given the existing theoretical results.

**Definition D.1** $((c, r)$-Approximate Nearest Neighbor). Given $n$ vectors $\mathsf{DB} = \{v_1, v_2, \ldots, v_n\} \in \mathbb{R}^{d \times n}$ and a distance function $\Delta(\cdot, \cdot)$, we say that an algorithm is a $(c, r)$-approximate nearest neighbor search algorithm if for any vector $q \in \mathbb{R}^d$ such that the true minimal distance between $q$ and any vector in $\mathsf{DB}$ is at most $r$, the algorithm outputs an index $i$ such that $\Delta(v_i, q) \leq c \cdot r$ with at least constant probability.

For the low-dimension case where $d = \Theta(\log n)$, we refer the reader to Prokhorenkova and Shekhovtsov (Prokhorenkova & Shekhovtsov, 2020) for the detailed discussion. Here, we will focus on the high-dimensional regime (also known as the sparse data case) such that the dimension $d = \omega(\log n)$, which is the most common setting for ANN search (recall that most embedding spaces are at least 100-dimensional).

Two notable theoretical results can be leveraged to analyze the graph-based ANN search. Laarhoven (Laarhoven, 2018) proves the following theorem:

**Theorem D.2** (Laarhoven (Laarhoven, 2018)). *Consider a database contains $n$ independently random vectors in the unit sphere in $\mathbb{S}^{d-1}$ and the distance is measured by Euclidean distance. For $c > 1$, there exists a $(c, r)$-graph-based ANN search algorithm with the $O(n^{1+\rho+o(1)})$ space complexity and $O(n^{\rho+o(1)})$ query time complexity while taking only $O(1)$ hops in the graph, where*

$$\rho \geq \frac{c^4}{2c^4 - 2c^2 + 1}.$$

We call this setting the *average case setting*. Intuitively, we can think about $n^\rho$ in the above theorem roughly denoting the average degree of the graph. For example, if we aim to have a 2-approximation ANN, then $\rho \approx 0.64$. We see that with the approximation factor $c$ getting worse, $\lim_{c \to \infty} \rho(c) = \frac{1}{2}$.

On the other hand, Diwan et al. (Diwan et al., 2024) studied the problem of building a navigable graph. On a high level, a navigable graph provides the guarantee for "in-distribution" queries for ANN search, that is, when the query vector will be exactly some vector in the database. They showed the following theorem:

**Theorem D.3** ((Diwan et al., 2024)). *For any $n$ vectors in $\mathbb{R}^d$, it is possible to build a navigable graph with average degree of at most $2\sqrt{n \ln n}$. Moreover, the "greedy-routing" strategy always succeeds in finding the correct vector in the database with at most 2 hops.*

The above two theorems provide the following theoretical insights: for the high-dimensional regime, to achieve a strong guarantee on the ANN search, the average degree of the graph will be roughly $n^{\frac{1}{2}+o(1)}$, and the query hop number will only be a constant. Therefore, if we use the same PIR technique to make the graph-based ANN search private, the underlying graph-information database is of size $n^{\frac{3}{2}+o(1)}$, where the total number of entries is $n$ and each entry is of size $n^{1/2+o(1)}$. It is interesting to see that this is not a typical setting of the PIR literature, because the entry size is much larger than a constant. If we plug in the best parameters of the client-preprocessing PIR (Zhou et al., 2024; Nguyen et al., 2024) into the Laarhoven's algorithm, we will get the following result:

**Theorem D.4.** *Assume the existence of one-way functions. For $c > 1$, there exists a $(c, r)$-graph-based private ANN search algorithm for Euclidean distance in the average case setting (random vectors on the unit sphere $\mathbb{S}^{d-1}$) with the following properties:*

- *Preprocessing cost: $\tilde{O}(n^{1+\rho+o(1)})$ communication and computation cost;*

- *Query cost:*

  - *$\tilde{O}(n^{1/2+\rho+o(1)})$ computation cost;*
  - *$\tilde{O}(n^{\rho+o(1)})$ communication cost;*

- *Storage cost:*

  - *Client: $\tilde{O}(n^{1/2})$;*
  - *Server: $\tilde{O}(n^{1+\rho+o(1)})$.*

*Here,*

$$\rho \geq \frac{c^4}{2c^4 - 2c^2 + 1}.$$

Specifically, we need to use the new result in Nguyen et al. (Nguyen et al., 2024) for optimizing the client storage. Notice that based on the existing client-specific preprocessing PIR lower bounds (Corrigan-Gibbs & Kogan, 2020; Persiano & Yeo, 2022; Larsen et al., 2020), to privately access a data structure of size $N$, the product between the client storage $S$ and the online time $T$ satisfies $S \times T = \Omega(N)$. In this sense, the above theoretical result is nearly tight in terms of the client storage and online query time if we follow the graph-based ANN paradigm: the client storage is $\tilde{O}(n^{1/2})$ and the online query time is $\tilde{O}(n^{1/2+\rho+o(1)})$, while the product matches the data structure size of $n^{1+\rho+o(1)}$.

**Gap between Theory and Practice.** Notably, the above theoretical results on graph-based ANN algorithms are based on the analysis of "high-degree" graphs, where the average degree is $\Omega(\sqrt{n})$, and the query hop number is a small constant. In practice, we see a different combination: popular graph-based ANN algorithm implementations usually pick a much smaller average degree, e.g., 32 or 64, while making the query hop number larger (e.g. scaling with the logarithm of the database size). It remains an interesting open question whether there exists a strong theoretical result for small-degree graphs with a search process invoking a large number of hops.

