# OpenReview forum: "Pacmann: Efficient Private Approximate Nearest Neighbor Search"
_ICLR.cc/2025/Conference — ICLR 2025 Poster_

### Official Review · Reviewer_BsSk · 2024-11-04

**Soundness:** 3
**Presentation:** 2
**Contribution:** 3
**Rating:** 6
**Confidence:** 2

**Summary:**

The paper introduces PACMANN, a private Approximate Nearest Neighbor (ANN) search scheme enabling clients to search vector databases without exposing their query vector to the server. Unlike previous methods relying on server-side encrypted searches, PACMANN offloads limited computation and storage to the client. It achieves up to 2.5× higher search accuracy on real-world datasets compared to prior private ANN schemes, approaching 90% of the quality of a state-of-the-art non-private ANN algorithm.

**Strengths:**

Query search mechanisms using fully homomorphic encryption (FHE) and multi-party computation (MPC) often suffer from high computational complexity. This paper addresses these challenges by introducing a localized search approach that performs iterative graph traversal on the client side. To minimize computation costs, it preprocesses Private Information Retrieval (PIR) to obtain private information from the server efficiently. Leveraging the Piano framework, the paper achieves a reduction in computation and communication costs to $O(\sqrt{n})$. For practical implementation, it further enhances efficiency with techniques such as beam search, fast start, and batched PIR queries.

**Weaknesses:**

The definition of privacy in the paper is somewhat vague. Could you clarify the attack model? Specifically, what are the attacker’s capabilities, and what information are they attempting to extract from the dataset? Providing these details would help make the definition of privacy more precise and formal.

In addition to FHE, MPC, and PIR, differential privacy (DP) can also provide formal privacy guarantees. While DP generally incurs low computational costs, it may reduce the model’s utility. Including a comparison with DP-related work would make the paper more comprehensive.

**Questions:**

NA

---

> ### Author Response · Authors · 2024-11-15
> **Author Response**
>
> Thank you for your comments!
>
>
>
> **Privacy Definition:** We will make our privacy definition clear in the next version. Specifically, we achieve cryptographically strong privacy assuming the adversary acting as the database owner is malicious, inheriting the privacy guarantee of the Piano PIR scheme: the adversary can choose the underlying database maliciously and respond to the client arbitrarily. It records all interactions with the client and tries to infer non-trivial information about the client’s queries. We prove that such an adversary learns negligible information about the queries. We do require an honest database owner for the utility guarantee – after all a malicious database owner can just refuse the service and respond randomly.
>
>  More formally, we proved the adversary’s views in the following two experiments are computationally indistinguishable:
>
> 	Real:
> 		The adversary chooses the underlying database. An honest client interacts with the adversary who acts as the server. In each time step, the adversary can adaptively choose a query vector based on its previous view, and the client will invoke the query algorithm with the chosen query vector as input. The adversary records the transcript of the interaction.
>
> 	Ideal:
> 		The adversary chooses the underlying database. A simulator simulates the client and interacts with the adversary who acts as the server. In each time step, the adversary can adaptively choose a query vector based on its previous view, and the simulator is invoked without the input query vectors. The adversary records the transcript of the interaction.
>
>
> **DP-guarantee**: A notable existing DP-based work is Wally[1], which implements the same cluster-based search functionality as Tiptoe. We achieved >90% recall@10 in the SIFT dataset, a significant improvement over the 35% recall@10 achieved by the cluster-based method used by Wally. Moreover, Wally needs additional waiting to batch many clients’ queries, since they offer a batched scheme. We will add this comparison in the final version.
>
> In terms of achieving DP-type privacy guarantee, replacing the underlying PIR in our scheme with DP-based PIR works, e.g., [2], could lead to further performance improvements. We leave this as an interesting future direction.
>
>
>
> [1]: Asi, Hilal, et al. "Scalable Private Search with Wally." arXiv preprint arXiv:2406.06761 (2024).
>
> [2]: Albab, Kinan Dak, et al. "Batched differentially private information retrieval." 31st USENIX Security Symposium USENIX Security, 2022.

---

### Official Review · Reviewer_4Hhv · 2024-11-04

**Soundness:** 3
**Presentation:** 4
**Contribution:** 3
**Rating:** 8
**Confidence:** 4

**Summary:**

This paper introduces PACMANN, a private approximate nearest neighbor (ANN) search method enabling clients to conduct privacy-preserving nearest neighbor queries across hundreds of millions of vectors. PACMANN utilizes state-of-the-art private information retrieval technique and customized searching algorithm to query vector database both privately and efficiently. PACMANN surpasses current leading private ANN search techniques in search quality and offers reduced latency for large-scale datasets. This approach could inspire more follow-ups applications in the area.

**Strengths:**

[S1] The motivation of this paper is clear and interesting, which is to address the privacy and efficiency challenges of remotely querying a vector database.

[S2] This paper successfully identifies the and addresses the limitations of existing approaches. It introduces a novel application of private information retrieval (PIR) to perform ANN search in a vector database. Compared with the baseline Tiptoe [1], this work does not involve heavy clustering-based ANN algorithm and homomorphic encryption, acheving promising efficiency for private query.

[1] Henzinger, Alexandra, et al. "Private web search with Tiptoe." Proceedings of the 29th symposium on operating systems principles. 2023.

[S3] This paper is well-organized with necessary background knowledge and clear illustration of high-level ideas. The language and presentation of this paper are easy understand.

[S4] This paper provides fair discussion on the proposed approach and identifies the limitation of this work. For example, it is more suitable for scenarios with good network connection, which would lead further research on potential following works on more general network connection.

[S5] This paper provides rigorous theoretical insights and illustrate the gap between theory and practice to enhance the understanding of graph-based ANN search.

**Weaknesses:**

[W1] The security analysis of the proposed approach is missing, which should be provided for a work specified in privacy-preservation. Although the core procedure PIR is invoked in black-box style, this paper should formally capture the potential information leakage during the workload. For example, the baseline of this paper Tiptoe provides a security analysis in Appendix D.

[W2] It would be better if the baseline Tiptoe can be implemented more completely in experiments. For example, this paper claims the efficiency limitation of Tiptoe comes from heavy homomorphic encryption and some other factors. As far as I know, Tiptoe just requires lightweight homomorphic encryption (i.e., homomorphic addition in LWE-style ciphertexts) and it is not that slow. For fair comparison, the simulation of Tiptoe should be equipped with homomorphic encryption to confirm the outperformance of this paper.

Typos:
“We could potentially the verifiable” in Appendix C.1.

**Questions:**

[Q1] Could you provide security analysis of the proposed method?

[Q2] It would be better to compare efficiency with Tiptoe-based solution.

---

> ### Author Response · Authors · 2024-11-15
> **Author Response**
>
> Thank you for your comments!
>
> **Q1**: We will make our security analysis clear in the next version. Specifically, we achieve cryptographically strong privacy assuming the adversary acting as the database owner is malicious, inheriting the security guarantee of the Piano PIR scheme: the adversary can choose the underlying database maliciously and respond to the client arbitrarily. It records all interactions with the client and tries to infer non-trivial information about the client’s queries. We prove that such an adversary learns negligible information about the queries. We do require an honest database owner for the utility guarantee – after all a malicious database owner can just refuse the service and respond randomly.
>
> **Q2**: Our current evaluation gave an *unfair* advantage to the Tiptoe-based solution by simulating its homomorphic encryption operations with simple plaintext operations. We still achieved performance gain in this case. To verify the validity of our simulation, we checked against the evaluation result in the original Tiptoe paper and our simulation was ~10% faster in online latency than their reported numbers. We document this comparison in Appendix D.2. We could not run the original Tiptoe implementation because it requires computationally intensive preprocessing, requiring dozens of GPU servers running hundreds of core-hours as documented in their paper. Our experiments are run on a single server-grade machine.

---

### Official Review · Reviewer_bC3L · 2024-11-05

**Soundness:** 3
**Presentation:** 3
**Contribution:** 2
**Rating:** 6
**Confidence:** 3

**Summary:**

This paper uses proposes a secure protocol for approximate nearest neighbours (ANN). The protocol is based on the client traversing a graph of the database held by a server, using a Private Information Retrieval (PIR) protocol to query each part of the graph that the client traverses. As the PIR protocol they use from previous work has O(sqrt(n)) online cost per query, and they need o(n) queries the latency is sublinear in the database size. Though the whole database must be sent during a preprocessing phase only a O(sqrt(n)) fraction of it need be stored by the client.

They provide some optimizations to the bsic idea, taking advantage of beam search to use the fact that it is cheaper per query to make several at once. they provide a small ablation study to verify this is infact helpful.

They then compare the quality of their results to the quality of some linear latency algorithm finding it improves on the baseline (on quality/latency tradeoff) when the database has size 2m to 50m depending on the network connection.

**Strengths:**

This is an interesting idea to explore.

The scheme works well asymptotically, being the first scheme with sublinear latency, thus constituting a theoretical breakthrough.

The idea is not just naively applied with thought gone into possibilities like beam search.

**Weaknesses:**

The breakthrough in asymptotics here is mostly coming from previous PIR work, though exploring the applicaiton of it is still valuable.

For the largest dataset they consider, after all their optimizations the reduction in latency is between 1.3x and 2.2x depending on the network (down from about 4 seconds)  to achieve this they require 3GB of client storage and 60GB of preprocesing communication. I am not aware of and they do not point to any realistic application in which this is a good tradeoff in practice.

**Questions:**

Are you aware of any realistic applications?

Shouldn't the expansion of the PACMANN acronym include the word nearest to explain the double N?

---

> ### Author Response · Authors · 2024-11-15
> **Author Response**
>
> Thank you for your comments!
>
> **Practical Applications**: Many practical scenarios that require privacy-preserving semantic information matching/searching could benefit from our solution, where the database sizes range from millions to billions. Here are some realistic examples:
> 1. Privacy-preserving harmful content detection. For example, a security software can utilize our solution to semantically match a suspicious file from a user against a list of potential viruses or harmful contents without compromising the user’s privacy. An example is [1] .
> 2. Privacy-preserving biometric information searching: we can perform fuzzy matching for the users’ sensitive biometric information against a public database for identification or information discovery. An example is [2].
>
> [1] https://www.apple.com/child-safety/pdf/CSAM_Detection_Technical_Summary.pdf
>
> [2] Hong, Matthew M., et al. "Secure Discovery of Genetic Relatives across Large-Scale and Distributed Genomic Datasets." International Conference on Research in Computational Molecular Biology. 2024.
>
>
> **Improvements over the previous work:**
> 1. We achieve the latency reduction and also a significant quality improvement simultaneously – improving from 35% recall@10 to 91% recall@10. If we aim for the same 35% recall, the latency reduction will be around 8-13x under different network conditions, as shown in Figure 2(b).
>
> 2. Our 1.3 - 2.2x latency reduction is w.r.t a *plaintext* linear scan that lower bounds the latency of the prior work, Tiptoe. Tiptoe needs a linear scan not in plaintext but in homomorphic evaluations. Even though they performed offline work to save online time, they still need quantization (full precision to 4-bit) to get reasonable online latency —- this is one source of their significant utility loss.
>
> 3. Even with strong utility improvement, our scheme is asymptotically better in computation than existing methods as we take a sublinear amortized cost per query, while the previous schemes require linear cost per query.
>
> We will also incorporate the editorial comments in the next version.

---

### Official Review · Reviewer_DbeZ · 2024-11-05

**Soundness:** 3
**Presentation:** 3
**Contribution:** 3
**Rating:** 6
**Confidence:** 2

**Summary:**

This paper presents Pacmann, a new private approximate nearest neighbor search scheme. The main claim is that this scheme is more accurate and efficient than prior private ANN search schemes.

The design of Pacmann includes a number of features, which I will try to recount here:
* Pre-process the search graph to ensure that every vertex in the graph has a bounded out-degree of $C$. Then, when one performs a greedy walk on the graph (toward finding an approximate neighbor), the search for the next hop needs to inspect $C$ vertices at most.
* To ensure privacy, the client will perform the search algorithm themselves. However, it is unreasonable to store the graph on the client side. If the client sends queries to the server in plain text, this may pose another privacy risk. The solution is a private information retrieval scheme, which allows the client to retrieve data from the server without revealing the index of interest. The paper appears to use an off-the-shelf PIR scheme from prior work, which, after client-side pre-processing, can achieve sublinear communication and computation costs for information retrieval.

A number of experiments are provided to support the effectiveness of the approach.

**Strengths:**

* The problem of study seems to be important, and the solution is natural and clearly written.
* The experiment results look promising.

**Weaknesses:**

(Disclaimer: I might not be an expert in critiquing this paper, as I don't have much experience in this area.)

* Can you comment on the scalability of your approach? I am quite concerned with the $O(\sqrt{n})$ communication and computation cost, as I understand that $n$ is the total number of vertices in the database. I see you ran your experiments up to 100M vertices. What does this scale mean in practice? I imagine, e.g., Google search would have a way larger scale of data than this.

**Questions:**

See above

**Details Of Ethics Concerns:**

As far as I can see, I don't see any ethics concern.

---

> ### Author Response · Authors · 2024-11-15
> **Author Response**
>
> Thank you for your comments!
>
> **Asymptotic Cost**: The $O(n)$ computation and communication are paid only once upfront during preprocessing and afterward every query takes only $O(CH\sqrt{n})$ computation and communication, where $C$ is the degree of the graph and $H$ is the maximum hop numbers for each query. The existing schemes such as Tiptoe[3] and Preco[4] pay $O(n)$ computation for every query. The $O(n)$ preprocessing computation is inevitable, implied by lower bounds for PIR (see [1][2]).
> Our scheme achieves $O(CH\sqrt{n})$ amortized communication, and the $\sqrt{n}$ part is necessary subject to using only symmetric key primitive and $\sqrt{n}$ client space, due to known lower bounds [5]. We can also implement our scheme with two non-colluding servers by having another server prepare the hints for the client during the preprocessing so that the preprocessing communication will be improved from $O(n)$ to $O(\sqrt n)$.
>
> **Practical Applications:** We are not expecting to scale our solution to a Google-search-size database, but many practical scenarios that require privacy-preserving semantic information matching/searching could still benefit from our solution, where the database sizes range from millions to billions. Here are some realistic examples:
> 1. Privacy-preserving harmful content detection. For example, a security software can utilize our solution to semantically match a suspicious file from a user against a list of potential viruses or harmful contents without compromising the user’s privacy [6].
> 2. Privacy-preserving biometric information searching: we can perform fuzzy matching for the users’ sensitive biometric information against a public database for identification or information discovery. See an existing example [7].
>
>
> [1] Beimel, Amos, Yuval Ishai, and Tal Malkin. "Reducing the servers computation in private information retrieval: PIR with preprocessing." Crypto, 2000.
>
> [2] Yeo, Kevin. "Lower bounds for (batch) PIR with private preprocessing." Eurocrypt, 2023.
>
> [3] Henzinger, Alexandra, et al. "Private web search with Tiptoe." SOSP, 2023.
>
> [4] Servan-Schreiber, Sacha, Simon Langowski, and Srinivas Devadas. "Private approximate nearest neighbor search with sublinear communication." IEEE S&P, 2022.
>
> [5] Ishai, Yuval, Elaine Shi, and Daniel Wichs. "PIR with Client-Side Preprocessing: Information-Theoretic Constructions and Lower Bounds." Crypto, 2024.
>
> [6] https://www.apple.com/child-safety/pdf/CSAM_Detection_Technical_Summary.pdf
>
> [7] Hong, Matthew M., et al. "Secure Discovery of Genetic Relatives across Large-Scale and Distributed Genomic Datasets." International Conference on Research in Computational Molecular Biology. 2024.

---

### Meta-Review · Area_Chair_9Lrk · 2024-12-20

**Metareview:**

This paper presents a new private Approximate Nearest Neighbor (ANN) search scheme named Pacmann that allows a client to perform ANN search in a vector database without revealing the query vector to the server.

Pacmann carefully offloads limited computation and storage to the client, achieving significantly better search quality than the state-of-the-art private ANN search schemes.

The reviewers agree that the problem of study is important and the solution is clearly written.

**Additional Comments On Reviewer Discussion:**

Some of the remaining concerns are:

a) The scalability of the approach and the O(\sqrt n) communication and computation cost.
b) The definition of privacy in the paper is somewhat vague.

---

### Decision · Program_Chairs · 2025-01-22

Accept (Poster)